# BLOCK-SPARSE RECURRENT NEURAL NETWORKS

## ABSTRACT

Recurrent Neural Networks (RNNs) are used in state-of-the-art models in domains such as speech recognition, machine translation, and language modelling. Sparsity is a technique to reduce compute and memory requirements of deep learning models. Sparse RNNs are easier to deploy on devices and high-end server processors. Even though sparse operations need less compute and memory relative to their dense counterparts, the speed-up observed by using sparse operations is less than expected on different hardware platforms. In order to address this issue, we investigate two different approaches to induce *block* sparsity in RNNs: pruning blocks of weights in a layer and using group lasso regularization with pruning to create blocks of weights with zeros. Using these techniques, we can create block-sparse RNNs with sparsity ranging from 80% to 90% with a small loss in accuracy. This technique allows us to reduce the model size by roughly $10\times$. Additionally, we can prune a larger dense network to recover this loss in accuracy while maintaining high block sparsity and reducing the overall parameter count. Our technique works with a variety of block sizes up to $32\times32$. Block-sparse RNNs eliminate overheads related to data storage and irregular memory accesses while increasing hardware efficiency compared to unstructured sparsity.

## 1 INTRODUCTION

Improvements in several applications such as speech recognition (Amodei et al., 2016), language modeling (Józefowicz et al., 2016), and machine translation (Wu et al., 2016) are a result of large Recurrent Neural Networks (RNNs) trained on large scale datasets. As the datasets available to train these models have grown, so have model sizes. Deployment of such large models is compute and memory intensive.

Pruning weights of deep neural networks is an effective strategy to reduce the overall memory and compute requirements of these models (Narang et al., 2017; Han et al., 2015). However, these approaches induce random, unstructured sparsity in the weight matrices. Speed-up obtained with unstructured sparsity on various hardware platforms are often lower than expected (as shown in Narang et al. (2017); Narang & Diamos (2017)). Sparse formats do not efficiently utilize the hardware resources due to storage overheads and irregular memory access. Block sparsity can address these issues. Saving indices of non-zero blocks instead of indices for non-zero elements reduces the storage overhead by a factor of block size. Block-sparse formats store blocks contiguously in memory reducing irregular memory accesses.

Another disadvantage of unstructured sparsity is that it cannot directly exploit array-data-paths in modern processors. These include the $16\times16$ TensorCore units in the Volta GPU (NVIDIA, 2017) or the $256\times256$ hardware units in the Tensor Processing Unit (TPU) (Jouppi et al., 2017). Structured sparsity in the form of two-dimensional blocks allows us to take advantage of these faster units.

In order to induce block sparsity in RNNs, we propose a block pruning approach that zeros out blocks of weights in the matrix while the network is training. At the end of training, the algorithm creates a block-sparse RNN. In addition to this pruning technique, we examine the efficacy of group lasso regularization (Yuan & Lin, 2006b) to induce block sparsity in the network. We also combine group lasso regularization with block pruning.

We demonstrate that block pruning and group lasso regularization with pruning are successful in creating block-sparse RNNs. Inducing block sparsity with $4\times4$ blocks in vanilla RNNs and Gated Recurrent Units (GRUs) (Cho et al., 2014) results in 9% to 17% loss in accuracy compared to the

dense baseline. Model size reduces by nearly 10×for speech recognition. Block sizes can be scaled up to 32×32 with our approach. We can also reduce accuracy loss by starting with a larger dense matrix than the baseline and then pruning it down while still reducing the number of parameters compared to the baseline. We demonstrate that this approach works with Long Short Term Memory (LSTM) (Hochreiter & Schmidhuber, 1997) cells for Language Modelling as well.

Our approach is agnostic to the optimization algorithm and does not require any hyper-parameter retuning (besides pruning and regularization hyper-parameters). Furthermore, since our approach does not require re-training the model, training time remains constant.

## 2 RELATED WORK

There have been several approaches to reduce the network size by pruning the model. Hanson & Pratt (1989) use several bias techniques to decay weights in a network. LeCun et al. (1989) and Hassibi et al. (1993) both use Hessian-based approaches to prune weights below a certain threshold. Simpler approaches like sorting or thresholding can be used to prune a neural network. Han et al. (2015) and Liu et al. (2015) prune Convolution Neural Networks (CNNs) while maintaining high accuracy. Yu et al. (2012) use a hard threshold to prune deep learning models. Narang et al. (2017) and Zhu & Gupta (2017) prune recurrent neural networks during the initial training run with a small accuracy loss using gradual pruning. Unlike our technique, all of the above approaches induce random, unstructured sparsity in neural networks.

Several approaches exist to induce structured sparsity in neural networks. Mao et al. (2017) use a simple threshold based technique to create structurally sparse CNNs. Yu et al. (2017) propose Scalpel, which prunes CNNs taking into account the underlying target hardware architecture. Wen et al. (2017) alter the structure of LSTMs to create cells with smaller memory footprint. They demonstrate that this technique works for language modeling on the Penn Tree Bank dataset. Our approach works with both vanilla RNN and GRU models trained on a large-scale datasets for speech recognition.

Regularization is a known method to induce sparsity in deep neural networks (Faraone et al., 2017; Fan et al., 2016). Group lasso regularization has been used as an efficient method for generating sparse structures (Yuan & Lin, 2006b; Kim & Xing, 2010). Wen et al. (2016) use group lasso regularization to induce structured sparsity in CNNs. Scardapane et al. (2017) also use group lasso regularization to induce sparisty in fully connected networks. To the best of our knowledge, none of these approaches have been used with RNNs trained on large-scale datasets.

Other approaches to reduce compute and memory footprint for deep learning models include quantization (Micikevicius et al., 2017; Vanhoucke et al., 2011; Rastegari et al., 2016; Gupta et al., 2015) and low-rank factorization (Denil et al., 2013; Denton et al., 2014). Our approach is orthogonal to these methods and can be combined with them.

## 3 IMPLEMENTATION

### 3.1 BLOCK PRUNING

Our approach to pruning deep learning models builds on the work by Narang et al. (2017). They propose a weight pruning algorithm that introduces random, unstructured sparsity in RNNs. In their work, they propose pruning weights below a monotonically increasing threshold. Their pruning strategy does not impose any structure on the weights.

We extend this approach to prune blocks of a matrix instead of individual weights. We divide the weight matrix into a grid of two-dimensional blocks with a fixed block size. Block size ranges between 4×4 to 32×32 in our experiments. In order to prune blocks, we pick the weight with the maximum magnitude to represent the entire block. If this maximum magnitude of a block is below the threshold, we set all the weights in that block to zeros. Figure 1 depicts the process of generating a block-sparse mask from a weight matrix for a given threshold. The block-sparse mask is multiplied with the weights to generate block-sparse weight matrix. The monotonically growing threshold ($\epsilon$) causes more blocks to be pruned as training progresses. We stop pruning more blocks after 40% of training has completed. All zeroed out blocks are held at zero until the end of training.

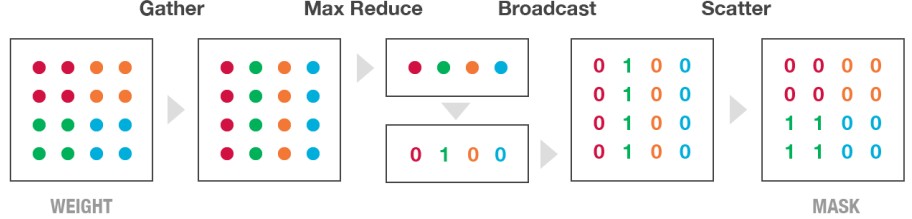

Figure 1: Generating block-sparse masks from a weight matrix

Table 1: Heuristics to pick hyper-parameters for block-pruning

| HYPER-PARAM | DESCRIPTION | HEURISTIC VALUES |
|---|---|---|
| *start_itr* | Iteration to start pruning | Start of second epoch |
| *ramp_itr* | Iteration to increase the rate of pruning | Start of 20% of total epochs |
| *end_itr* | Iteration to stop pruning more parameters | Start of 40% of total epochs |
| *start_slope* ($\theta$) | Initial rate of increasing the threshold | See Equation 2 |
| *ramp_slope* ($\phi$) | Rate of increasing threshold after ramp iteration | $1.2\theta$ to $1.7\theta$ |
| *freq* | Number of iterations after which $\epsilon$ is updated | 100 |

Narang et al. (2017) use six hyper-parameters to determine the threshold at a given iteration. Table 1 provides the description and heuristics (adapted for block pruning) for these hyper-parameters. The *start_slope* and *ramp_slope* determine the rate at which the threshold increases. In order to determine *start_slope*, they recommend using weights from an existing dense model. To achieve 90% sparsity, they assign $q$ to the weight that is the 90th percentile of the absolute values in a weight matrix. Assuming $\phi$ is $1.5\theta$, they use Equation 1 to determine $\theta$.

$$\theta = \frac{2 \times q \times freq}{2 \times (ramp\_itr - start\_itr) + 3 \times (end\_itr - ramp\_itr)} \tag{1}$$

For block pruning, we need to modify the *start_slope* to take into account the number of elements in a block ($N_b$). In order to calculate the *start_slope*, we first calculate *start_slope* for weight pruning ($\theta_w$) using the Equation 1. Given $\theta_w$, we suggest using Equation 2 to determine the initial slope ($\theta_b$) for block pruning. Based on empirical results, we have found that using this approach allows us to achieve block sparsity ranging from 85% to 95%. Further tuning of these hyper-parameters is required to achieve desired block sparsity.

$$\theta_b = \theta_w \times \sqrt[4]{N_b} \tag{2}$$

We prune all the recurrent and fully connected layers in the network using the same block size. The pruning hyper-parameters are same for each type of layer in the network — recurrent weight layer and linear or fully connected layer.

## 3.2 GROUP LASSO REGULARIZATION

Group lasso is a type of weight regularization that works on groups of weights. For each group, we add a loss term proportional to the $\ell_2$ norm of the group.

$$L = L_{\text{training}} + \lambda_g \sum_{g=1}^{G} \|w^{(g)}\|_2$$

where $w^{(g)}$ is a group of weights, $\|w^{(g)}\|_2$ is the $\ell_2$ norm of the group, and $G$ is the total number of groups. Our use of $\ell_2$ norm is a variant of the more general group lasso defined in Yuan & Lin (2006a) as $\|n\|_K = (n'Kn)^{1/2}$.

A large enough $\lambda_g$ will drive all weights within certain groups to zeros. The choice of grouping varies by application, and group lasso is widely-used to induce various kinds of structured sparsity (Wen et al., 2017; Scardapane et al., 2017). By choosing groups to exactly match our 2D grid of blocks, we can induce block-sparsity. Thus, group lasso is an existing sparsity algorithm that we can readily compare to our block pruning approach.

In addition, we extend group lasso to work with block pruning. The groups match the 2D grid of blocks used by the pruning algorithm. One interpretation of weight regularization is that less important weights are driven towards zero and more important weights retain large absolute values. Thus, group lasso guides the selection of blocks to prune. We apply group lasso to coincide with the pruning schedule. We use a relatively small $\lambda_g$ to avoid underfitting due to excessive regularization. We turn off group lasso when the pruning schedule ends, which is typically after around 40% of training epochs. Weights that were already set to zero remain unchanged after this point.

Group lasso is related to the well-known $\ell_1$ regularization. In Appendix A, we discuss exploration of $\ell_1$ regularization combined with weight pruning.

## 4 EXPERIMENTS

We present results on two different applications: Speech Recognition (Section 4.1) and Language Modelling (Section 4.2).

### 4.1 SPEECH RECOGNITION

We run block sparsity experiments on two different speech recognition models from Amodei et al. (2016). The RNN model consists of seven bidirectional vanilla recurrent layers with 1760 hidden units for a total of 67 million parameters. The GRU model consists of three recurrent layers with GRU cells with 2560 hidden units for a total of 115 million parameters. Both models use the Connectionist Temporal Classification (CTC) (Graves et al., 2006) cost function. We use a training set of 2100 hours of speech and validation set of 3.46 hours. The Character Error Rate (CER) results are reported on an independent test set, consisting of 2.9 hours of English data.

In order to introduce block sparsity in these models, we run two different types of experiments — Block Pruning (BP) and Group Lasso with block pruning (GLP). We prune weights in the recurrent layers (both linear and recurrent weights) and fully connected layers. Biases, batch-normalization parameters and weights in the convolutional and CTC layers are not pruned since they account for a small portion of the total weights in the network. No existing hyper-parameter changes were required for sparse training runs. The models are trained using Nesterov Stochastic Gradient Descent (SGD) with momentum. All models are trained for 25 epochs.

In Section 4.1.1, we report results for different sparse models pruned with 4×4 blocks and compare these results with other pruning approaches. In Section 4.1.2, we discuss the impact of varying the block size on the accuracy of the model.

### 4.1.1 BLOCK SPARSITY

Initially, we prune the dense RNN model. Using BP, we are able to reduce the parameter count for both these models by nearly 10×. As shown in Table 2, the block-sparse RNN model with 1760 hidden units has an overall block sparsity of 89% with a CER of 17.93.

As mentioned in Section 3, group lasso by itself can induce block-sparsity. However, as shown in Table 2, group lasso results in significantly worse CER than our block pruning approach. In order to achieve high sparsity (80% or higher) with group lasso, we need to set $\lambda_g$ to a relatively high value. This high regularization factor hurts the model accuracy. The dense baseline model is trained without any regularization. Therefore, group lasso results in underfitting the training data due to the high value of $\lambda_g$. Group lasso could be more successful in inducing block-sparsity where the dense model overfits the training dataset.

Table 2: Bidirectional RNN model results. Block-Sparse models are trained with 4x4 blocks

| MODEL | LAYER SIZE | # PARAMS (in millions) | CER (% LOSS) | EPOCHS | PRUNING ALGORITHM |
|---|---|---|---|---|---|
| RNN Dense | 1760 | 67 | 15.36 (0.0%) | 25 | N/A |
| RNN Block-Sparse | 1760 | 10.9 | 30.14 (-96%) | 25 | Group lasso |
| RNN Sparse | 1760 | 8.3 | 18.91 (-23%) | 25 | Yu et al. (2012) |
| RNN Sparse | 1760 | 7.3 | 17.32 (-13%) | 25 | Narang et al. (2017) |
| RNN Sparse | 1760 | 7.1 | **15.41 (-0.3%)** | **60** | Han et al. (2015) |
| RNN Block-Sparse | 1760 | 7.3 | 17.93 (-17%) | 25 | Ours (BP) |
| RNN Block-Sparse | 2560 | 12.9 | **15.89 (-3.4%)** | 25 | Ours (GLP) |
| RNN Block-Sparse | 3072 | 25.8 | **15.66 (-1.9%)** | 25 | Ours (BP) |

Table 3: GRU model results with $4 \times 4$ blocks

| MODEL | LAYER SIZE | # PARAMS (in millions) | CER (% LOSS) | EPOCHS | PRUNING ALGORITHM |
|---|---|---|---|---|---|
| GRU Dense | 2560 | 115 | 15.42 (0.0%) | 25 | N/A |
| GRU Block-Sparse | 2560 | 10.8 | **16.78 (-8.8%)** | 25 | Ours (GLP) |
| GRU Block-Sparse | 3584 | 25.6 | **16.23 (-5.3%)** | 25 | Ours (BP) |

**Comparison to other pruning methods:** In addition to group lasso, we compare our block pruning approach with three existing pruning methods. As shown in Table 2, our block-sparse model achieves better accuracy than the hard thresholding scheme in Yu et al. (2012). Sparse RNNs generated using Narang et al. (2017) is about 4% better than the block-sparse model. The sparse RNN model generated using iterative pruning (Han et al., 2015) is significantly better than than block-sparse model. However, this approach requires training the model for 60 epochs instead of 25 epochs for all other approaches. This results in 180 hours of additional training time for the RNN model. This 2-3×increase in training time may not be practical for state-of-the-art models trained on large datasets, which usually need weeks of training time. Additionally, all the above approaches generate random, unstructured sparsity in the model. In current hardware, the compute and memory savings with block sparsity are significantly higher than random sparsity. We discuss the performance aspect in more detail in Section 5.

**Larger models:** In order to recover the accuracy loss with our approach, we train sparse models with more hidden units in each recurrent layers. For RNN models, we increase the hidden layer size to 2560 and 3072. As shown in Table 2, the RNN sparse 3072 is only 1.9% worse than the dense baseline model. The 2560 and 3072 sparse RNN models reduce the overall parameter count by 5×and 2.5×respectively relative to the dense model with 1760 hidden units in each layer.

**GRU model:** Similar to the RNN models, the block-sparse GRU model can reduce the overall parameter count by 11×. As shown in Table 3, the block-sparse GRU model achieves slightly higher sparsity (90%) with a CER of 16.23 which is only 9% worse than the dense GRU model. This indicates that the block-sparse GRU model retains most of the capacity of the dense model. As demonstrated with the RNN model, pruning a larger GRU model with 3584 hidden nodes reduces the accuracy loss to about 5% while still shrinking the model by 4.5×relative to the dense model with 2560 hidden nodes in each layer.

### 4.1.2 BLOCK SIZE VARIATION

Table 4 shows that block pruning works for block sizes upto $32 \times 32$. Increasing the block size to $16 \times 16$ and $32 \times 32$ requires reducing the sparsity to 83.6% and 79.1% respectively for RNN models

Table 4: Results for GRU model with 2560 layer size and bidirectional RNN model with 170 layer size pruned with different block sizes using BP.

| MODEL | BLOCK SIZE | # PARAMS (in millions) | SPARSITY | CER (% LOSS) |
|---|---|---|---|---|
| RNN Block-Sparse | 4x4 | 7.3 | 89.2% | 17.93 (-17%) |
| RNN Block-Sparse | 12x2 | 10.8 | 84.1% | 16.96 (-10%) |
| RNN Block-Sparse | 8x8 | 10.7 | 84.1% | 17.66 (-15%) |
| RNN Block-Sparse | **16x16** | 11.1 | 83.6% | 17.10 (-11%) |
| RNN Block-Sparse | **32x32** | 14.1 | 79.1% | 16.67 (-8.5%) |
| | | | | |
| GRU Block-Sparse | 4x4 | 16.2 | 86.0% | 16.97 (-10%) |
| GRU Block-Sparse | **16x16** | 20.8 | 81.9% | 16.84 (-9.2%) |

Table 5: Word language modelling results on Penn Tree Bank using BP for 4×4 blocks.

| MODEL | LAYER SIZE | # PARAMS (in millions) | PERPLEXITY (% LOSS) | EPOCHS | PRUNING ALGORITHM |
|---|---|---|---|---|---|
| LSTM Dense | 1500 | 66.0 | 78.29 (0.0%) | 55 | N/A |
| | | | | | |
| LSTM Block-Sparse | 1500 | 23.1 | **77.04 (1.6%)** | 55 | Ours (BP) |
| LSTM Block-Sparse | 1500 | 11.6 | 80.25 (-2.5%) | 55 | Ours (BP) |
| LSTM Block-Sparse | 1500 | **7.95** | 82.72 (-5.7%) | 55 | Ours (BP) |

to obtain good accuracy. Similar results hold true for the GRU model as well. Large sparse blocks reduce memory overhead for storing non zero values and can take advantage of array data-paths in modern processors. Therefore, even though large blocks achieve lower sparsity, they result in lower memory and compute requirements.

The exact choice of block size for a given application depends on the underlying hardware used for inference. For example, NVIDIA's Volta processor supports 16×16 blocks whereas ARM processors support blocks of 12x2. We demonstrate that our approach is agnostic to block size and can be used to generate block-sparse models for arbitrary blocks.

### 4.2 NEURAL LANGUAGE MODELLING

We conducted block pruning experiments on Penn Tree Bank (PTB) (Marcus et al., 1993) dataset using word level language models. For our experiments, we use the large LSTM model with 1500 hidden units from Zaremba et al. (2014). The hyperparameters are unchanged from the original model, except for slightly increased dropout keep probability which ranges from 0.4 to 0.52 for the sparse models. We prune weights in the embedding, LSTM and softmax layers of the model.

We report results on the test set using BP with 4×4 blocks in Table 5. Block pruning can reduce the parameter count by nearly 3×while retaining the accuracy of the dense model. With a 5% loss in accuracy, we can reduce the parameter count by 8.3×. There is a trade-off between sparsity and accuracy of the model. For inference, we would pick the model that meets the desired memory and compute budget. Further work remains in evaluating this technique for large scale datasets like the Billion word datasets (Chelba et al., 2013) for language modelling.

## 5 PERFORMANCE

Sparse formats incur at least three types of overhead: i) indexing overhead, ii) irregular memory accesses, and ii) incompatibility with array-data-paths, all of which are mitigated by using larger block sizes.

**Indexing Overheads**. Sparse formats use extra memory to track the location of each non-zero value. For example, the compressed-sparse-row (CSR) format uses between one and two extra index values for each non-zero value. Assuming that neural network weights and indices are represented with 16-bits as in Micikevicius et al. (2017), this is at least $100\%$ overhead. Block sparsity reduces this overhead by a factor of the block size because the index is shared over the entire block. For example, using a block size of 4x4 reduces the memory bloat to $6.25\%$, and using a block size of 16x16 reduces it to less than $1\%$.

**Irregular Memory Accesses**. Caches lines, DRAM row buffers, and TLBs provide the best performance when memory is accessed in relatively large contiguous units (e.g. 64 bytes for cache lines, 4KB for a DRAM row) as opposed to in fine-grained random accesses. Block-sparse formats store blocks contiguously in memory, resulting in large coalesced accesses.

**Array Data-Paths**. Block-sparse models make it easier to exploit array-data-paths in modern processors. There are significant advantages of using these units, for example, on the Volta V100 GPU, they enable up to 8x higher throughput than the SIMD data-paths. In order to keep these units busy, the block size should be at least as large as the hardware data-path size ($16\times16$ or larger on V100).

## 5.1 INFERENCE PERFORMANCE

Inference performance depends of three different factors: accuracy, latency of evaluation and memory requirements. In order to understand the trade-off between unstructured sparsity and block sparsity, we benchmark the General Matrix Multiply (GEMM) speed-up and memory reduction for a single layer in the speech recognition RNN model. We evaluate GEMM speed-up with batch size of 16 using NVIDIA's CuSparse and CuBLAS libraries on a TitanX Maxwell GPU. Sparse matrices are represented in CSR or Block-CSR format depending on the sparsity structure. Memory savings are calculated using CSR and Block Sparse Row (BSR) from Scipy module in Python. We evaluate a single layer with different block sizes. We also evaluate the best unstructured sparsity result obtained using iterative pruning from Han et al. (2015).

As shown in Table 6, the unstructured sparsity model (Han et al., 2015) achieves the best accuracy but requires much longer training time and does not improve the compute time relative to the dense model. For a small loss in accuracy, block-sparse models can significantly reduce both compute and memory requirements. For example, layers with 16x16 block sparsity reduce memory consumption by $11\times$ and speedup compute by $3\times$ with a 10% loss in accuracy. Additionally, Figure 2 shows that block-sparse matrices achieve higher speed-up than unstructured sparsity for large batch sizes for RNN and GRU layers. The speed-up is achieved due to reducing irregular memory accesses and improving load balance. Future work involves efficient implementation of block-sparse kernels to take advantage of array-data-paths in modern processors.

Table 6: Accuracy, speed-up and memory reduction for sparse layers. Block-sparse layers achieve higher speedup and memory reduction with some loss in accuracy.

| MODEL | LAYER SIZE | GEMM SPEEDUP | MEMORY SAVINGS | CER (% LOSS) | ALGORITHM |
|---|---|---|---|---|---|
| RNN Sparse | 1760 | $1.0\times$ | $5.0\times$ | **15.41 (-0.3%)** | Han et al. (2015) |
| RNN Block-Sparse | 2560 | $1.1\times$ | $6.2\times$ | 15.89 (-3.4%) | BP (4x4) |
| RNN Block-Sparse | 1760 | $1.5\times$ | $7.1\times$ | 16.67 (-8.5%) | BP (32x32) |
| RNN Block-Sparse | 1760 | **$3.0\times$** | $11\times$ | 17.10 (-11%) | BP (16x16) |
| RNN Block-Sparse | 1760 | $1.9\times$ | **$17\times$** | 17.93 (-17%) | BP (4x4) |

## 6 IMPACT OF SPARSITY ON ACCURACY

Using our baseline RNN model, we run many Weight Pruning (WP) (using (Narang et al., 2017)) and block pruning experiments, varying hyper-parameters to produce a spectrum of results ranging from 70% to 97% sparsity. For these experiments, the models are trained for 20 epochs and the accuracy is reported on the validation set. As shown in Figure 3, models pruned using WP with

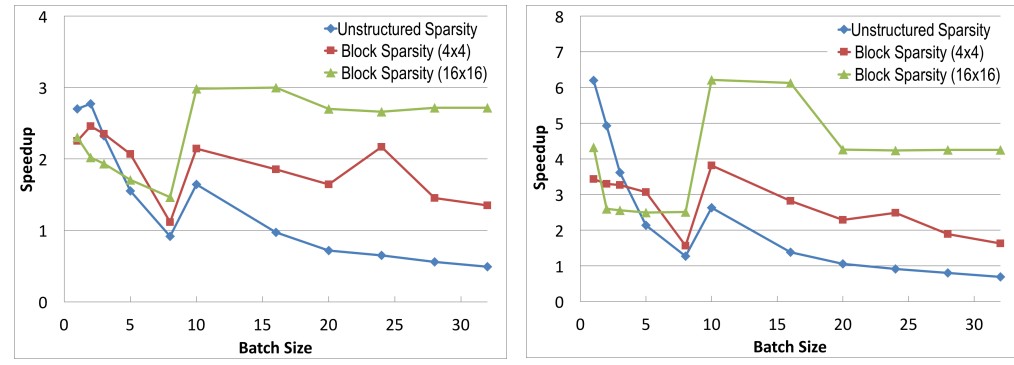

(a) Speed-up for RNN 1760 layer matrix multiply    (b) Speed-up for GRU 2560 layer matrix multiply

Figure 2: Speed-up for sparse matrix multiply over GEMM. RNN matrix sizes are (1760,1760) with 90% sparsity and (1760, batch_size). GRU matrix sizes are (7680,2560) with 95% sparsity and (2560, batch_size). Block-sparse matrices achieve consistently good speedup across batch-sizes

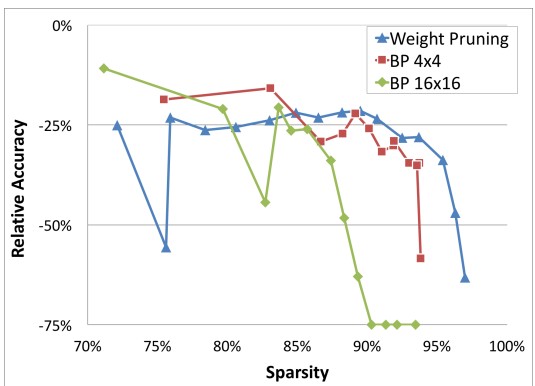

Figure 3: Relative accuracy for different block sizes (4x4, 16x16) and WP for varying sparsity on the RNN 1760 model. Any models with relative accuracy worse than -75% are capped at 75%.

sparsity less than 95% have relative accuracy ranging from -20% to -27%. Increasing the sparsity for the model beyond 95% results in 30% or more accuracy loss. This "accuracy cliff" is earlier for models pruned with block sparsity. For block size 4×4, models with sparsity greater 90% yield a relative accuracy loss of 30% or higher. Similarly, for blocks of 16×16, models with sparsity greater than 86% have 30% or more accuracy loss. A similar trend is observed for block size 32×32. This indicates that there is a trade-off between sparsity and block-size for a given accuracy. Larger blocks reach the "accuracy cliff" sooner.

## 7 CONCLUSION AND FUTURE WORK

We have demonstrated that using block pruning and group lasso combined with pruning during training can build block-sparse RNNs that are about as accurate as the dense baseline models. The block-sparse models have significantly fewer parameters than the dense baselines reducing memory requirements. Block-sparse models can take advantage of the underlying hardware efficiently.

We would like to investigate if pruning can be performed even earlier in the training, thereby allowing us to train sparse models. Training sparse models would allow us to reap the benefits of sparsity during training resulting in lesser compute and memory demands. Further work remains to implement efficient block-sparse matrix multiplies for array-data-paths in modern processors that would provide increased speed-up during deployment.

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

## A $\ell_1$ AND $\ell_{1/2}$ REGULARIZATION

Prior to our work with group lasso regularization, we considered $\ell_1$ and $\ell_{1/2}$ regularizers to induce sparsity in the network. These regularizers act on individual weights and could aid in inducing unstructured sparsity in the network. $\ell_1$ regularization is defined as:

$$L = L_{\text{training}} + \lambda \sum_{i=1}^{k} |w_i|$$

where $|w_i|$ is the absolute value of a weight and $k$ is the total number of weights. Note the gradient expression for each weight $w_j$:

$$\frac{\partial}{\partial w_j} \sum_{i=1}^{k} |w_i| = sgn(w_j)$$

As with the group lasso experiments described in 3.2, we explore $\ell_1$ regularization with and without pruning. The weight pruning (WP) algorithm from Narang et al. (2017) is used along with regularization. The motivation is the same as group lasso block sparsity experiments: either to guide pruning or to produce sparsity directly.

We also explore $\ell_{1/2}$ regularization which is defined as:

$$L = L_{\text{training}} + \lambda \sum_{i=1}^{k} |w_i|^{1/2}$$

Fan et al. (2016) uses $\ell_{1/2}$ regularization to produce sparsity directly. The gradient for $\ell_{1/2}$ regularization is $\frac{1}{2}|w_j|^{-1/2}$. This term is smaller for weights with larger magnitude. Our expectation is that $\ell_{1/2}$ will drive unimportant weights towards zero while leaving large weights relatively unaffected, thus avoiding the accuracy loss associated with excessive regularization.

For our $\ell_1$ and $\ell_{1/2}$ experiments, we use the Deep Speech 2 Bidirectional RNN baseline model described in Section 4. These models are trained for 25 epochs on our internal training dataset of 2000 hours. The results are reported on a independent test set consisting of 2.9 hours.

Table 7: $\ell_1$ and $\ell_{1/2}$ results with the bidirectional RNN model with 1760 hidden units

| MODEL | # PARAMS (in millions) | SPARSITY | CER | RELATIVE PERF | PRUNING ALGORITHM |
|---|---|---|---|---|---|
| RNN Dense | 67 | 0.0% | 15.36 | 0.0% | N/A |
| RNN Sparse | 7.3 | 89.2% | **17.32** | -12.8% | Weight pruning |
| RNN Sparse | 11.2 | 83.6% | 24.8 | -61.5% | $\ell_1$ |
| RNN Sparse | 7.4 | 89.1% | **17.28** | -12.5% | $\ell_1$ with pruning |
| RNN Sparse | 6.6 | 90.3% | 18.50 | -20.4% | $\ell_{1/2}$ with pruning |

Without pruning, $\ell_1$ model results in significantly worse accuracy compared to the dense baseline. Combining $\ell_1$ with weight pruning allows us to recover the loss in accuracy with similar sparsity. The $\ell_{1/2}$ with pruning model performs worse than the $\ell_1$ with pruning model. Comparing the two regularizers, this result indicates that $\ell_1$ is better at guiding pruning than $\ell_{1/2}$, more suitable as a regularizer, or both.

Similar to group lasso experiments, $\ell_1$ regularization experiments require a significantly higher $\lambda$ to achieve high sparsity without any pruning. We suspect that these regularizers would be more successful in inducing sparsity for models that overfit the training training dataset.

# B PRUNING CHARACTERISTICS

In this section, we discuss some pruning characteristics and how they relate to training and accuracy of the models.

## B.1 PRUNING SCHEDULE

In Figure 4a, we plot the pruning schedule of a recurrent and linear layer of the bidirectional model trained with Block Pruning (BP) and Weight Pruning (WP) (Narang et al., 2017) and Group lasso with block pruning (GLP). For all three algorithms, pruning begins just after the first epoch at 2700 iterations. The BP and GLP models result in a sharper curve with more weights being set to zero in a short span of iterations. In these experiments, we use the *max* function to reduce the blocks to a single value which could be the cause of the sharpness in pruning. Also the GLP model reaches 90% sparsity just before 10,000 iterations which is significantly earlier than the BP model. GLP training encourages sparsity early on in the training run by pushing the blocks of weights towards zero.

## B.2 OUTPUT CONNECTIONS

Figure 4b shows the histogram of the number of output connections for all the neurons in a network for two models with different sparsity pruned with BP. The 94% sparse model does significantly worse than the 89% sparse. For the model with 89% sparsity, only 180 neurons have all their output weights set to zero out of a total of 38270. This model produced good accuracy relative to the dense baseline. However, increasing the sparsity to 94% for the layer results in 1620 neurons having all zero output weights. Additionally, a lot more neurons have a smaller number of non-zero output weights.

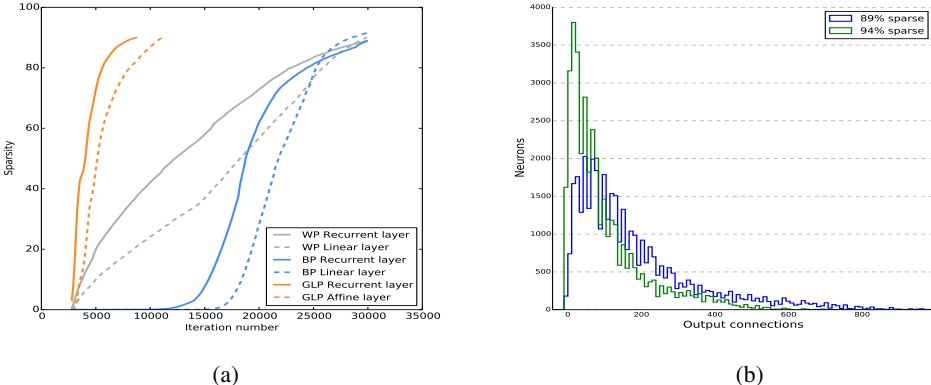

(a)    (b)

Figure 4: Figure 4a shows the pruning schedule for two layers in the network for WP, GLP and BP models. The GLP and BP models use block size of 4x4. Figure 4b plots the histogram of the number of output connections for all neurons in the network using block pruning with 4×4 blocks.

## B.3 SPARSITY VS LAYERS

Figure 5 shows the sparsity of all the recurrent layers in the network using BP and WP. All recurrent layers have the same pruning hyper-parameters. Layer 1 is the first recurrent layer and layer 14 is the final recurrent layer before the CTC cost layer. For both block pruning and weight pruning, we see that the initial layers are pruned more aggressively compared to the final layers. Increasing sparsity in the layers closer to the output results in poor accuracy. Additionally, the variance in sparsity across the layers increases with the block size. This increasing variance makes it harder to increase the block size beyond 32×32 with the same pruning hyper-parameters for all recurrent layers.

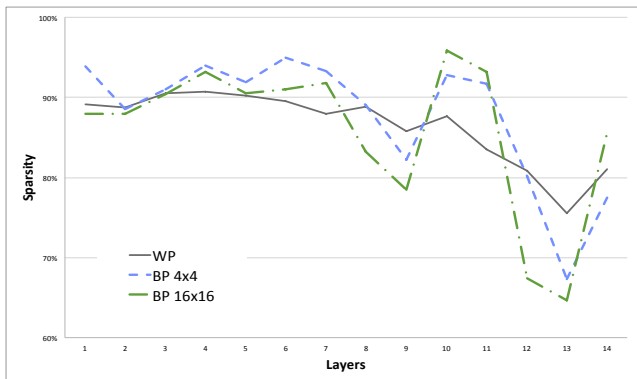

Figure 5: Sparsity of different recurrent layers in the network in the RNN model, pruned using BP and WP.

