# OpenReview forum: "Block-Sparse Recurrent Neural Networks"
_ICLR.cc/2018/Conference — Reject_

### Official Review · AnonReviewer3 · 2017-11-21
**Good start, but needs better comparison against existing work.**

**Rating:** 5
**Confidence:** 3

**Review:**

The authors propose a block sparsity pruning approach to compress RNNs. There are several ways. One is using  group LASSO to promote sparsity. The other is to prune, but with a very specialized schedule as to the pruning and pruning weight, motivated by the work of Narang et al 2017 for non-group sparsity.  The block sizes used in experiments are about 4x4, 8x8, up to 32 x 32. The relative performance degradation ranges between 10% to 96%, depending on the method, severity of compression, and task. The speedup for a matrix multiply is between 1.5x to 4x, and varies according to batch size.

This is certainly a well-motivated problem, and the procedure is simple but makes sense. Also, the paper contains a good overview of related work in compression, and is not hiding anything.  One paper that is missing is

Scardapane, S., Comminiello, D., Hussain, A., & Uncini, A. (2017). Group sparse regularization for deep neural networks. Neurocomputing, 241, 81-89.

A major complaint is the lack of comparison of results against other compression techniques. Since it is a block sparsity approach, and the caching / fetching overhead is reduced, one does not need to have competitively superior results to basic pruning approaches, but one should come close on the same types of problems. This is not well presented. Additionally, the speedup should be superior to the sparse methods, which is also not shown (against previously published results, not personally run experiments.)

Another issue I find is the general writing, especially for the results section, is not entirely clear. For example, when showing a relative performance degradation of 96%, why is that happening? Is that significant? What should an implementer be aware of in order to avoid that?

Finally, a meta issue to address is, if the block size is small (realistically, less than 64 x 64) usually I doubt there will be significant speedup. (4x is not considered terribly significant.) What we need to see is what happens when, say, block size is 256 x 256? What is the performance degradation? If you can give 10x speedup in the feedforward part (testing only) then if you have a 10% degradation in performance that might be acceptable in certain applications.

Overall, I believe this is a very promising and well-motivated work, but needs to be "marinated" further to be publishable. Actually, with the inclusion of 2-3 tables against known, previously published results, and clearly stated benefits, I would change my review to accept.

Minor complaints:

The axis labels/numbers in figure 2 are too small.

Also, please reread for some grammar / writing issues (last paragraph of 1st page, caption of figure 2, for example)

I think also figure 2 should be rerun with more trials. The noise in the curves are not showing a highly interpretable trend. (Though, actually, batch size = 8 being super low might have a significance; can this be explained?)

---

> ### Author Response · Authors · 2017-12-15
> **Response to Reviewer 3**
>
> We thank the reviewer for the helpful feedback and comments. Thank you for pointing out the missing reference. We have added it to the paper and will update it along with rest of the changes.
>
> We agree that we should compare our results to more prior approaches. In the paper, we have compared the block pruning approaches to prior pruning approaches like Narang et. al. In our paper, Group Lasso only experiments are also another baseline. However, this isn't clear in the paper. We are working on adding comparisons to more prior work including Han et al. However, all prior work induces random unstructured sparsity in the network and therefore the speedup is limited. We will add these baselines in our paper and clearly state the benefits of our approach.
>
> Group Lasso alone doesn't work for our models. Without regularization, the baseline dense model doesn't overfit the dataset. Therefore, adding regularization to generate sparse blocks hurts the accuracy of the model. We suspect that group lasso regularization is resulting in underfitting. Therefore, we don't advocate using Group Lasso alone to introduce block sparsity when training on large datasets. Instead, we recommend using block pruning or group lasso combined with block pruning which produces good accuracy results.
>
> It is true that for certain processors (like a Tensor Processing Unit) larger block sizes like 256x256 would realize higher speedup. However, for GPUs and processors with SIMD units, smaller block sizes can result in higher speedup with efficient kernels. For example, ARM CPUs support 16x1 vectors and the NVIDIA Volta TensorCores support 16x16 blocks. For 4x4 and 16x16 blocks, we show that the speedup ranges from 4.5x to 1.5x depending on the matrix size and minibatch size using libraries that were not tuned explicitly for block sparse RNNs. Recently, new block sparse kernel libraries have been released (https://blog.openai.com/block-sparse-gpu-kernels/) which demonstrate that it is possible to achieve speedup close to the theoretical maximum of 1(1-s/100) where s is the sparsity of the network. The work shows that smaller blocks of 32x32 even with 80% sparsity can achieve about 4x speedup.
>
> While benchmarking speedups, we run each GEMM 10,000 times. We will run the benchmark with more iterations to ensure that the speedup holds true. Additionally, we will run with more minibatch sizes to confirm the trend. Also, at batch size of 8, the dense kernels are very fast and therefore speedup observed with sparse kernels is small. For batch sizes larger than 8, the dense time increases significantly resulting in improved speedup.
>
> We will update the paper fixing axis labels and reread for grammar issues. Thanks again for the review and feedback.

---

> > ### Comment · AnonReviewer3 · 2017-12-19
> > **Re block size experiments**
> >
> > I guess what I'm looking for is a performance vs speedup plot comparison, parametrized by different block sizes. If the plot looks "knee-like", e.g. you can get a lot of speedup for not that much performance hit, then the experiments will be consistent with the motivation. But I think it will be hard to motivate this method if your block size is limited to 8x8; in general people are fine with waiting 2x for an experiment if it means not rewriting an entire numerical library.

---

> > > ### Comment · AnonReviewer3 · 2018-01-07
> > > **response to revision**
> > >
> > > After reading the revision I am still unable to change to accept.
> > >
> > > 1) The main thing, although there are added comparisons (against Han and Yu, etc) none of them seem particularly close to what the paper is trying to do. On the other hand, Mao and Wen, which seem to have more block-like structured sparsity, is not compared. After skimming the Yu paper, it seems that they are also using block sparsity, but any comparison between this paper and Yu is limited to a performance metric in which Yu outperforms this paper, and no runtime  or memory usage comparison is given. So though there is more comparison, the comparisons added are still not entirely fair.
> > >
> > > 2) It is still not clear to me that 32 x 32 block size is big enough. Actually, table 6 is a bit strange, as there is no clear correlation between speedup and blocksize. I accept that there exist hardware that can exploit 16 x 16 block sparsity, but the claim that this paper’s algorithm is agnostic to block size is not convincing.
> > >
> > > 3) Table 2: the comparison against Han, what is the performance of Han at 25 epochs? The claim that Han is better but requires more epochs isn’t really solid without that datapoint.
> > >
> > > 4) Figures 2,3 still seem a bit messy. Are they done over many trials?
> > >
> > > 5) The explanation that group LASSO is underperforming because it is underfitting is not very satisfying. Both penalty and thresholding form of model complexity reduction have the same overfitting and underfitting tradeoffs, so it is not convincing that there does not exist a scheme under which group penalty regularization cannot do as well as thresholding. Of course the exact mechanism isn’t the same, so there may actually be a performance superiority for thresholding (even if all lambda choices are swept) but here should be a different explanation for this.
> > >
> > > 6) (minor) lasso —> LASSO

---

> > > > ### Author Response · Authors · 2018-01-09
> > > > **Response**
> > > >
> > > > Thank you for the quick response to the new revision. Here are some thoughts and responses to your questions.
> > > >
> > > > 1. Comparisons to other baselines
> > > >
> > > > Han et al.
> > > >
> > > > We cite https://arxiv.org/abs/1510.00149. Both Han's approach and ours look to reduce neural network compute and memory requirements by producing sparsity in weight matrices, so we feel comparison is relevant. Han's approach offers greater accuracy at the expense of longer training time and other compute/memory drawbacks, as detailed in Table 6 in our paper.
> > > >
> > > > Yu et al.
> > > >
> > > > The work by Yu et al. focusses on pruning fully connected layers during training. It's similar to Han et. al. since they keep the top k weights at given point training. However, it doesn't increase the training time and uses a hard threshold instead of a gradual pruning approach (Narang et. al). Therefore, we added this comparison to a paper to give another datapoint with fixed number of epochs. We demonstrate that the our block pruning approach achieves better accuracy with fewer parameters than Yu et al.
> > > >
> > > > >>>>After skimming the Yu paper, it seems that they are also using block sparsity, but any comparison between this paper and Yu is limited to a performance metric in which Yu outperforms this paper, and no runtime or memory usage comparison is given
> > > >
> > > > In their paper, Yu et al. propose a different data structure to store sparse weights, however, the sparsity enforced is still unstructured and random. Their new data structure (described in Section 3.3 of the paper), achieves slightly more memory savings (6.6x v/s 5x for 90% sparsity for the RNN layer) than the CSR matrix format. However, since an implementation of this format isn't available, we don't think it's fair to add this to the paper.
> > > >
> > > > In Table 6, we picked the unstructured sparsity model with the best accuracy and computed memory and runtime savings for it. We didn't include results from Yu et al. or Narang et al. in Table 6 since they induce random sparsity and have worse accuracy than Han et al. Please let us know if we have missed something.
> > > >
> > > > Mao et al.
> > > >
> > > > We cite https://arxiv.org/pdf/1705.08922. We agree that comparison to Mao et al would be valuable. Their paper analyzes various approaches to structured sparsity, and their vector pruning approach in particular could be applied to speech recognition RNN models.
> > > >
> > > > Wen et al.
> > > >
> > > > We cite https://arxiv.org/abs/1608.03665. They focus on sparsity in convolutional layers, so their approach isn't directly applicable to RNN models. We also cite https://arxiv.org/abs/1709.05027. This paper is fairly new (mid-Sep 2017 as compared to our late Oct submission date). We agree that a comparison of this approach applied to speech recognition RNN models would be valuable. In addition, for this approach, we can directly compare weight count and accuracy for neural language modeling results on the Penn Tree Bank dataset:
> > > >
> > > > # Parameters Perplexity (% Loss) Algorithm
> > > >
> > > > 66.0M 78.29 N/A
> > > >
> > > > 25.2M 76.03 (+2.9%) Wen2017
> > > >
> > > > 21.8M 78.65 (-0.5%) Wen2017
> > > >
> > > > 23.1M 77.04 (+1.6%) Ours (BP)
> > > >
> > > > 11.6M 80.25 (-2.5%) Ours (BP)
> > > >
> > > > 7.95M 82.72 (-5.7%) Ours (BP)
> > > >
> > > > We didn't include this comparison in our paper because it's not entirely fair: in Wen et al's experiments, they only induce sparsity in the RNN and softmax layers, whereas we also induce sparsity in the embedding layer.
> > > >
> > > >
> > > > 2. Memory & Compute savings for different block sizes
> > > >
> > > > Table 6 highlights the speedup, memory compression and accuracy for different models with different block sizes. For brevity, we omitted the sparsity of the model from the table. This information is available in Tables 2 and 4. Since the 16x16 model is about 5% more sparse than the 32x32 model, it achieves superior speedup and compression. Additionally, we're hoping that our work will motivate people to develop efficient block-sparse kernels for modern hardware like Volta with array datapaths.
> > > >
> > > > > the claim that this paper’s algorithm is agnostic to block size is not convincing.
> > > >
> > > > We can modify this claim in the paper to state that our approach works for block sizes up to 32x32.

---

> > > > ### Author Response · Authors · 2018-01-09
> > > > **Response continued**
> > > >
> > > > Continuing the response from the previous comment.
> > > >
> > > > 3. Accuracy of Han et al. at 25 epochs
> > > >
> > > > The work Han et. al involves pruning a pre-trained model and re-training it 2 or 3 times to achieve accuracy close to the original baseline. In our work, we prune a pre-trained speech recognition model (trained for 20 epochs) with varying sparsity (75%, 85% and 90%) and retrain it. The accuracy of these pruned models after 25 epochs are 10%, 23% and 30% respectively worse than the dense models. However, training the model for longer allows the model to recover the loss in accuracy. We decided to report the 60 epoch accuracy since this closely resembles the original work published in Han et al.
> > > >
> > > > In addition, the work by Yu et. al involves pruning the smallest k weights at a specific epoch during training. This is similar to the work by Han et. al. but it doesn't increase model training by using a hard threshold during the first training iteration. We demonstrate that our gradual block pruning approach outperforms this hard thresholding approach for fixed number of training epochs.
> > > >
> > > > 4. Speedup figures
> > > >
> > > > The speedup figures are run for 100,000 trials which is 10x more than the trials with the previous version of the paper. This should be sufficient to eliminate any noise in the performance benchmarking.
> > > >
> > > > 5. Group lasso discussion
> > > >
> > > > We don't broadly claim that block-pruning outperforms group lasso regularization in general. But we are confident in our results for these specific speech recognition RNN models. These baseline models were SOTA at the time of their publication and they don't employ any L1, L2, dropout, or dropconnect. These models don't benefit from such regularization strategies.
> > > >
> > > >
> > > > Some of our analysis of this issue was omitted from the paper for brevity. For example, we analyzed the distribution of nonzero weight magnitudes in our baseline, block pruning (BP), group lasso (GL), and group-lasso-with-pruning (GLP) runs. Across baseline, BP, and GLP runs, the distributions looked similar. GL runs' distributions, conversely, were more concentrated near zero. This seem to indicate that group lasso reduces the weight magnitude of all the groups of weights in the network including non-zero ones. This led to our underfitting hypothesis for the group lasso runs. On the other hand, the block pruning approach allows the other blocks to remain unconstrained and weights can have large magnitude. This may be the reason why block pruning performs better than group lasso. Future work of the paper involves diagnosing the exact cause of poor accuracy with group lasso regularization.
> > > >
> > > >
> > > > Thanks again for taking the time to review our paper.

---

### Official Review · AnonReviewer2 · 2017-11-22
**The paper proposes to 0 out blocks of weights while training RNNs and further aid the process by utilizing group lasso regularization. As demonstrated empirically, the learned networks are sparse and can be run efficiently while showing minimal loss of accuracy.**

**Rating:** 7
**Confidence:** 4

**Review:**

Compressing/pruning of neural networks is required to enable running on devices with limited compute resources. While previous works have proposed to 0 out weights, especially for the case of RNNs, in an unstructured way, the current paper proposes to 0 out weights blocks at a time via thresholding. The process is further aided by utilizing group lasso regularization. The resulting networks are sparse, memory efficient and can be run more efficiently while resulting in minimal loss in accuracy when compared to networks learned with full density. The proposed techniques are evaluated on RNNs for speech recognition and benefits clearly spelled out. Further experiments thresh out how much benefit is provided by thresholding (block sparsity) and regularizing via group lasso.

The paper quality seems high, presentation clarity sufficient, the ideas presented (especially the use of group lasso) well thought out and original, and the work seems significant. If I were to nitpick then I would suggest trying out these techniques on RNNs meant for something other than speech recognition (machine translation perhaps?).

---

> ### Author Response · Authors · 2017-12-15
> **Response to Reviewer 2**
>
> Thank you for your review and feedback. We are working on extending this approach to Language Modelling and will hopefully have results on small dataset soon. We will update the paper with the results if they are available before the rebuttal deadline.

---

### Official Review · AnonReviewer1 · 2017-11-26
**Methods for inducing sparsity in terms of blocks of weights in neural networks.**

**Rating:** 5
**Confidence:** 4

**Review:**

Thanks to the authors for their response.

Though the paper presents an interesting approach, but it relies heavily on heuristics (such as those mentioned in the initial review) without a thorough investigation of scenarios in which this might not work. Also, it might be helpful to investigate if there ways to better group the variables for group  lasso regularization. The paper therefore needs further improvements towards following a more principled approach.

=====================================
This paper presents methods for inducing sparsity in terms of blocks of weights in neural networks which aims to combine benefits of sparsity and faster access based on computing architectures. This is achieved by pruning blocks of weights and group lasso regularization. It is demonstrated empirically that model size can be reduced by upto 10 times with some loss in prediction accuracy.

Though the paper presents some interesting evaluations on the impact of block based sparsity in RNNs, some of the shortcomings of the paper seem to be :

- The approach taken consists of several heuristics rather than following a more principled approach such as taking the maximum of the weights in a block to represent that block and stop pruning till 40% training has been achieved. Also, the algorithm for computing the pruning threshold is based on a new set of hyper-parameters. It is not clear under what conditions the above settings will (not) work.

 - For the group lasso method, since there are many ways to group the variable, it is not clear how the variables are grouped. Is there a reasoning behind a particular grouping of the variables. Individually, group lasso does not seem to work, and gives much worse results. The reasons for worse performance could be investigated. It is possible that important weights are in different groups, and group sparsity is forcing some of them to be zero, and hence leading to worse results. It would be insightful to explain the kind of solver used for group lasso regularization, and if that works for large-scale problems.

 - The results for various kinds of sparsity are unclear in the sense that it is not clear how to set the block size a-priori for having minimum reduction in accuracy and still significant sparsity without having to repeat the process for various choices.

Overall, the paper does not seem to present novel ideas, and is mainly focused on evaluating the impact of block-based sparsity instead of weight pruning by Han etal. As mentioned in Section 2, regularization has been used earlier to achieve sparsity in deep networks. In this view the significance over existing work is relatively narrow, and no explicit comparison with existing methods is provided. It is possible that an existing method leads to pruning method such as by Han etal. leads to 8x decrease in model size while retaining the accuracy, while the proposed method leads to 10x decrease while also decreasing the accuracy by 10%. Scenarios like these need to be evaluated to understand the impact of the method proposed in this paper.

---

> ### Author Response · Authors · 2017-12-15
> **Response to Reviewer 1**
>
> We thank the reviewer for their comments and helpful feedback.
>
> We present several heuristics related to hyperparameters, and we regard these heuristics as an aid for practitioners, to narrow the range of their hyperparameter search. We agree with the reviewer that it's not clear under what conditions these heuristics might break down. The requirement for hyperparameter tuning is a drawback of our approach, but other pruning approaches within the field share this drawback.
>
> For our group lasso experiments, we pick groups to exactly match the blocks in our block pruning experiments. This is a regular tiling of 2D blocks across an individual 2D weight matrix. Unlike some prior work using group lasso, our reasoning for this grouping is not based on any expectation about grouping or correlation in the input features. Instead, we choose this grouping purely to induce a block sparsity format in the weight matrix that is efficient for hardware implementation. We'll update the paper to clarify these points.
>
> Group Lasso alone doesn't work for our models due to underfitting. Without regularization, the baseline dense model doesn't overfit the dataset. Therefore, adding regularization to generate sparse blocks hurts the accuracy of the model. Group Lasso could be more effective in inducing block sparsity in a data-limited problem.
>
> In section 4.3, we demonstrate that our block pruning approach works for block sizes up to 32x32. The section demonstrates a tradeoff between block size and sparsity. The exact choice of block sizes will depend on the underlying hardware. E.g. The best block size for the Nvidia Tesla V100 is 16x16 due to the array data paths used by the TensorCores. We will add some notes to the paper to aid a practitioner in making this choice.
>
> Finally, we are working on adding comparisons to previous work including Han et. al. and will update the paper with these results including the pros and cons of each approach.
>
> Our work is novel because this is first approach to introduce block sparsity for Recurrent Neural Networks with vanilla RNN and GRU cells. To the best of our knowledge, no prior work has applied Group Lasso Regularization to large RNN models to induce block sparsity. Additionally, our block pruning algorithm does not increase training time unlike some prior work.

---

### Author Response · Authors · 2018-01-05
**New Revision of the paper**

We have submitted a new revision to the paper. The changes (as requested by the reviewers) include:

- Comparison to other baselines (including Han et. al.)
- Results on Neural Language Modeling with the Penn Tree Bank dataset
- Inference performance section to highlight the benefits of block sparsity over unstructured sparsity
- Speedup with more repeats and batch sizes. Also, added results from NVIDIA's block - CSR format for block-sparse layers
- Add motivation for the choice of blocks for group lasso
- Other grammatical and stylistic fixes
- Moved some of the discussion section to the appendix

We thank the reviewers for their consideration and helpful feedback.

---

### Decision · Program_Chairs · 2018-01-29
**ICLR 2018 Conference Acceptance Decision**

**Decision:**

Reject

**Comment:**

Pros
-- Interesting approach to induce sparsity, trains faster than alternative approaches
Cons
-- Fairly complex set of heuristics for pruning weights
-- Han et al. works well, although the authors claim it takes more time to train, which may not not hold for all training sets and doesn’t seem like a strong enough reason to choose an alternative appraoch

Given these comments, the AC recommends that the paper be rejected.